# Molecular feature-based classification of retroperitoneal liposarcoma: a prospective cohort study

**Mengmeng Xiao[1,2†], Xiangji Li[2,3†], Fanqin Bu[3†], Shixiang Ma[2], Xiaohan Yang[3], Jun Chen[2], Yu Zhao[3], Ferdinando Cananzi[4], Chenghua Luo[1,2]\*, Li Min[3]\***

[1]Department of Retroperitoneal Tumor Surgery, Peking University People's Hospital, Beijing, China; [2]Department of Retroperitoneal Tumor Surgery, Peking University International Hospital, Beijing, China; [3]Department of Gastroenterology, Beijing Friendship Hospital, Capital Medical University, State Key Laboratory for Digestive Health, National Clinical Research Center for Digestive Disease, Beijing Digestive Disease Center, Beijing Key Laboratory for Precancerous Lesion of Digestive Disease, Beijing, China; [4]Department of Biomedical Sciences, Humanitas University, Milan, Italy

## eLife Assessment

This study presents a **valuable** and simplified classification system for predicting clinical outcomes of RPLS patients. The data were collected and analyzed using **solid** and validated methodology and can be used as a starting point for personalized treatment of RPLS. The work will be of interest to scientists working in the field of RPLS.

**\*For correspondence:**
pkuihlch@163.com (CL);
minli@ccmu.edu.cn (LM)

[†]These authors contributed equally to this work

**Competing interest:** The authors declare that no competing interests exist.

## Abstract

**Background:** Retroperitoneal liposarcoma (RPLS) is a critical malignant disease with various clinical outcomes. However, the molecular heterogeneity of RPLS was poorly elucidated, and few biomarkers were proposed to monitor its progression.

**Methods:** RNA sequencing was performed on a training cohort of 88 RPLS patients to identify dysregulated genes and pathways using clusterProfiler. The GSVA algorithm was utilized to assess signaling pathway levels in each sample, and unsupervised clustering was employed to distinguish RPLS subtypes. Differentially expressed genes (DEGs) between RPLS subtypes were identified to construct a simplified dichotomous clustering via nonnegative matrix factorization. The feasibility of this classification was validated in a separate validation cohort (n=241) using immunohistochemistry (IHC) from the REtroperitoneal SArcoma Registry (RESAR). The study is registered with https://clinicaltrials.gov/ under number NCT03838718.

**Results:** Cell cycle, DNA damage and repair, and metabolism were identified as the most aberrant biological processes in RPLS, enabling the division of RPLS patients into two distinct subtypes with unique molecular signatures, tumor microenvironment, clinical features, and outcomes (overall survival [OS] and disease-free survival [DFS]). A simplified RPLS classification based on representative biomarkers (LEP and PTTG1) demonstrated high accuracy (area under the curve [AUC]>0.99), with patients classified as LEP+ and PTTG1-, showing lower aggressive pathological composition ratio and fewer surgery times, along with better OS (HR = 0.41, p<0.001) and DFS (HR = 0.60, p=0.005).

**Conclusions:** Our study provided an ever-largest gene expression landscape of RPLS and established an IHC-based molecular classification that was clinically relevant and cost-effective for guiding treatment decisions.

**Funding:** This work was supported by grants from the Beijing Municipal Science and Technology Project (Z191100006619081), National Natural Science Foundation of China (82073390), and Young Elite Scientists Sponsorship Program (2023QNRC001). The study sponsors had no role in the design and preparation of this manuscript.
**Clinical trial number:** NCT03838718

## Introduction

Retroperitoneal liposarcoma (RPLS) is a soft tissue sarcoma originating in the retroperitoneum with an insidious onset. Traditional surgical resection has been regarded as a primary and curable treatment strategy of RPLS for the past 50 years (*Ecker et al., 2016*). However, the anatomical complexity and biological properties of sarcoma brought great difficulty in achieving microscopically margin-negative resection, leading to a high postoperative recurrence rate in RPLS patients. During the past decade, scientists tried to improve the postoperative survival of RPLS patients by personalized surgical resection and neoadjuvant/adjuvant therapies, but the effect was not satisfactory (*Littau et al., 2021*; *Gronchi et al., 2015*; *Gronchi et al., 2009*; *Gronchi and Pollock, 2013*; *Pisters, 2009*).

Recently, precision medicine greatly enriched the therapeutic approaches and reformed the clinical decision-making chain of tumor diagnosis and treatment, prolonging the median survival of main tumor types 2–10 times (*Kam et al., 2021*; *Zeng and Jin, 2022*; *Alifrangis et al., 2019*; *Frese et al., 2021*). Biomarker-based patient stratification and targeted therapy together make up the kernel of precision medicine, which is intrinsically based on the molecular profiling of cancers. However, our knowledge of the molecular features of RPLS is limited, and few clinically applicable molecular biomarkers and targeted drugs are available for RPLS treatment. Only sporadic molecules such as CDK4 (*Pilotti et al., 2000*), MDM2 (*Binh et al., 2005*), AURK4 (*Yen et al., 2019*), and CCNDBP1 (*Yang et al., 2021*) have been reported as prognostic and diagnostic biomarkers, but these biomarkers were poorly represented and verified. Therefore, it is crucial to reveal the molecular landscape of RPLS and explore a feasible classification for its diagnosis and treatment.

Here, we conducted a comprehensive investigation into the molecular characteristics of RPLS through the delineation of the largest gene expression landscape ever assembled for this rare disease entity. By identifying both RPLS-specific genes and prognostic biomarkers, we unveiled their intricate relationships with clinical parameters. Our findings revealed the existence of two distinct molecular subtypes within all RPLS patients, characterized by diverse pathological compositions, enriched signaling pathways, and varying clinical outcomes. This highlights the limitations of relying solely on traditional pathological classification for surgical decision-making in certain cases where patients exhibit favorable histological features but poor prognoses, emphasizing the pivotal role of molecular subtyping in guiding individualized treatment strategies and enhancing patient management. To facilitate practical application in clinical settings, we developed a simplified RPLS classification system based on key biomarkers (LEP and PTTG1) representative of each subtype. Notably, this classification scheme was validated in a larger cohort of RPLS patients through immunohistochemistry (IHC) assays (*Figure 1*), laying the groundwork for precise surgical interventions guided by molecular insights in the realm of RPLS treatment.

## Materials and methods
### Patients and tissue specimens

Patients who were diagnosed with RPLS amenable to surgical resection were eligible for the study. The RPLS histology was confirmed according to the WHO criteria done on biopsy or surgical specimen by a dedicated sarcoma pathologist. Exclusion criteria comprised age <18 years, severe psychiatric disorders impairing informed consent or study compliance, and inability to ensure adequate follow-up. Tumor specimens from 88 RPLS patients (training cohort 1, *Supplementary file 1*; training cohort 2, *Supplementary file 2*) and another cohort of 241 RPLS patients (validation cohort, *Supplementary file 3*) were obtained from our local hospital. These cohorts are sourced from REtroperitoneal SArcoma Registry (RESAR, NCT03838718). All the patients underwent curative resection from January 2015 to May 2019. RPLS tissue specimens were snap-frozen in liquid nitrogen within 1 hr and

**eLife digest** Retroperitoneal liposarcoma (or RPLS for short) is a rare type of cancer that forms a tumor deep in the abdomen. Although it can be treated through surgery, results vary widely, and the cancer often returns. Determining which RPLS tumours are likely to respond well to treatment, and which are more aggressive, can be challenging. This is partly because many of the tumors appear similar under a microscope, making it difficult to identify those that might require more intense treatment.

For some types of cancer, researchers have started to overcome this problem by identifying molecular markers – specific genes, proteins and other molecules – that are unique to tumor cells. This allows for a better understanding of tumor behavior and helps pinpoint which treatments are most likely to be successful. However, the relevant molecular markers for RPLS remain unknown, preventing this precision medicine approach from being applied to this type of cancer.

To help close this gap, Xiao et al. studied the genetic activity of tumors from 88 RPLS patients to look for activity associated with more aggressive tumors. Analysis revealed two distinct groups of tumors. One group showed high activity of a gene known as *LEP*, which is involved in metabolism. Tumors with high *LEP* activity were less aggressive, with patients surviving longer and requiring fewer surgeries. The other tumor group showed high activity of a gene known as *PTTG1*, which plays a role in cell division and DNA repair. When overactive, these processes can make tumors grow and spread more quickly. In this group, tumors were more likely to return after surgery. Based on these findings, Xiao et al. developed a simple test using the two genes to classify tumors. When tested in a larger group of 241 patients, this method successfully sorted tumors into the two groups with more than 99% accuracy.

The findings of Xiao et al. suggest that the activity of *LEP* and *PTTG1* could potentially be used as a molecular marker for identifying RPLS patients that might benefit from more intensive treatments. However, further research is required to confirm how reliable these markers are before they can be used to diagnose patients in the clinic.

then stored in a –80°C refrigerator before use. Clinical information was collected from the medical records, and no patient had undergone previous chemotherapy or radiotherapy. Overall survival (OS) was defined as the interval between the latest surgery and death from tumors or between the latest surgery and the last observation taken for surviving patients. Disease-free survival (DFS) was defined as the interval between the latest surgery and diagnosis of relapse or death. Informed consent for surgical procedures and specimen collection was obtained from each patient. This study has been reported in line with the REMARK criteria (*McShane et al., 2005*). The data of the external validation cohort (GSE30929) was downloaded from GEO (https://www.ncbi.nlm.nih.gov/geo/).

## RNA sequencing, primary data processing, and analysis

Total RNA was extracted from training cohort 1 (*Supplementary file 4*) and training cohort 2 (*Supplementary file 2*) using TRIzol Reagent (Invitrogen). RNA degradation and contamination were monitored with 1% agarose gel. RNA purity was checked by the NanoPhotometer spectrophotometer (IMPLEN, Los Angeles, CA, USA). RNA concentration was measured using the Qubit RNA Assay Kit with the Qubit 2.0 Fluorometer (Life Technologies, CA, USA). RNA integrity was assessed using the RNA Nano 6000 Assay Kit of the Agilent Bioanalyzer 2100 System (Agilent Technologies, CA, USA).

A total amount of 3–5 µg RNA per sample was used as input material for the RNA library. Sequencing libraries were generated using NEBNext Multiplex Small RNA Library Prep Set for Illumina (NEB, USA) following the manufacturer's recommendations, and index codes were added to attribute sequences to each sample. The clustering of the index-coded samples was performed on a cBot Cluster Generation System using TruSeq SR Cluster Kit v3-cBot-HS (Illumina). After cluster generation, the strand-specific cDNA was sequenced on an Illumina NovaSeq 6000 platform, and single-end reads were generated (Novogene Bioinformatic Technology, Beijing, China).

FPKMs of mRNAs and noncoding RNAs in each sample were calculated by Cuffdiff (v2.1.1). FPKMs were calculated based on the length of the fragments and read counts mapped to this fragment. These

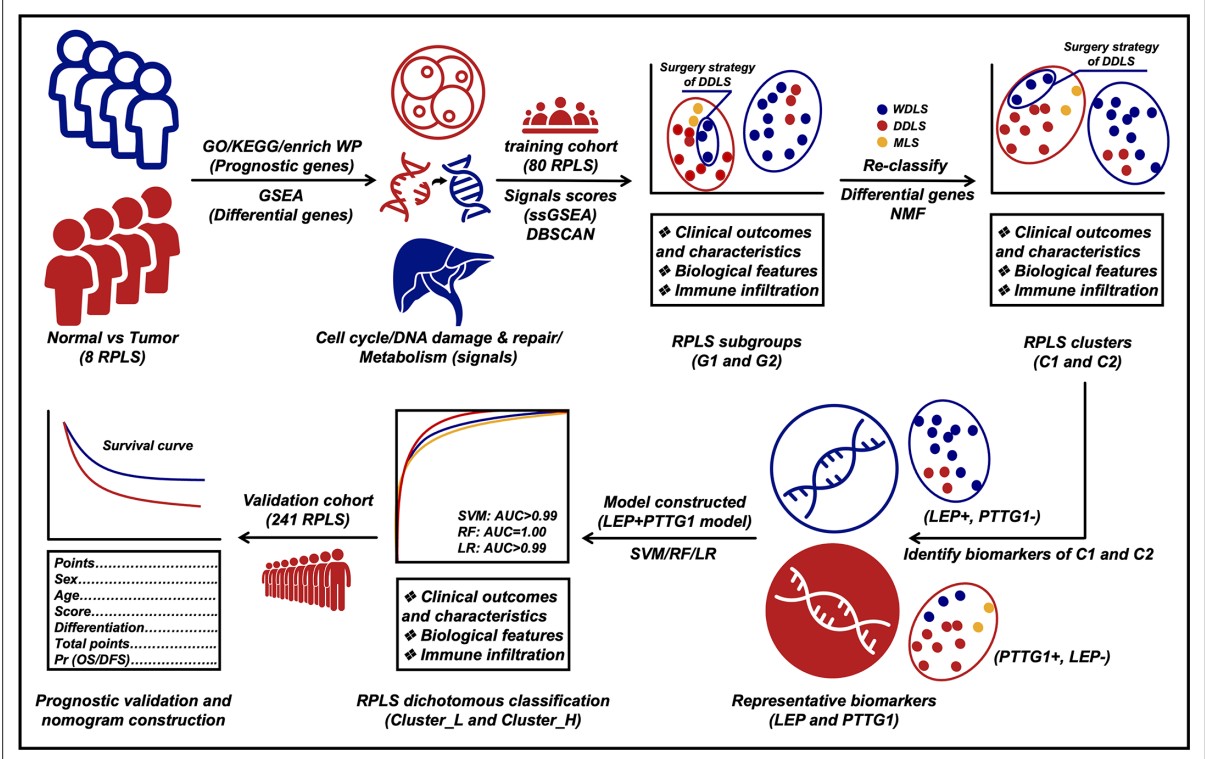

**Figure 1.** Flow diagram of exploring retroperitoneal liposarcoma (RPLS) dichotomous classification.

sequencing data have been deposited at the Open Archive for Miscellaneous Data (OMIX) database of China National Center for Bioinformation (CNCB) under the accession number OMIX002786.

## Identification of differential genes

Gene difference analysis was performed to determine the differential genes (differentially expressed genes [DEGs]). An adjusted false discovery rate (FDR)<0.05 and |log2FC|>0.585 were considered significant. This process was conducted with the R package 'limma'.

## Identification of prognostic genes

Cox univariate regression analysis was used to screen the prognostic genes of RPLS. Results of p<0.05 were considered significant. This process was conducted with the R package 'survival'.

## GSEA and immune infiltrate analysis

Gene set enrichment analysis (GSEA) was performed in the tumor and normal groups to explore the biological signaling pathways. Pathway annotation files were downloaded from the MSigDB (https://www.gsea-msigdb.org/gsea/index.jsp) platform. This process was conducted by the GSEA R package to elucidate the representative HALLMARK and REACTOME pathways enriched in RPLS patients. Immunocyte infiltration (immune score and stromal score) was measured by the Estimation of STromal and Immune cells in Malignant Tumor tissues using Expression data (ESTIMATE) algorithm. This process was completed via the 'estimate' R package.

## Functional annotation

Functional enrichment analyses were performed to elucidate the possible biological processes and signaling pathways of the prognostic genes. Gene Ontology (GO) and Kyoto Encyclopedia of Gene and Genomes (KEGG) analyses were conducted by the R package 'clusterProfiler', and the FDR<0.05 was considered significantly enriched.

## Consensus clustering with t-SNE

After evaluating the relative abundance level of related pathways, the Euclidean distance was calculated between any two samples and condensed into two-dimensional points using t-distributed stochastic neighbor embedding (t-SNE) (*Guo et al., 2019a*) and subsequently visualized automatically with the density-based spatial clustering of applications with noise (DBSCAN) algorithm. This consensus clustering was conducted with the R packages 'Rtsne' and 'dbscan'.

## Consensus clustering with NMF

Nonnegative matrix factorization (NMF) was used to perform RPLS subtyping. Specifically, NMF was applied to gene expression matrix *A* which contained gene sets of major signaling pathways and prognostic genes. Matrix *A* was factorized into two nonnegative matrices *W* and *H*. Repeated factorization of matrix *A* was performed, and its outputs were aggregated to obtain consensus clustering of RPLS samples. The optimal number of subtypes was selected according to cophenetic, dispersion, and silhouette coefficients. This consensus clustering was conducted with the R package 'NMF'.

## Construction of machine learning models

Machine learning models based on biomarkers were constructed by logistic regression (LR), support vector machine (SVM), and random forest (RF). These models were specifically tailored to analyze biomarker data in order to predict clinical outcomes in surgical patients.

LR is a statistical method that establishes a relationship between a set of independent variables and a binary outcome. It calculates the probability of an event occurring based on the six to seven input features derived from biomarkers relevant to the surgical patient. SVM is a supervised learning algorithm that categorizes data points by identifying the optimal hyperplane that separates distinct classes within a high-dimensional space. This approach effectively maps biomarker data into a multi-dimensional space to facilitate accurate classification of patient outcomes. RF is an ensemble learning technique that generates multiple decision trees during training and aggregates the results to make predictions. By leveraging this method, we can enhance predictive accuracy by mitigating overfitting and increasing model robustness when analyzing biomarker-driven patient data.

The performance of these machine learning models was assessed using the area under the curve (AUC) metric. A higher AUC value indicates superior discriminatory power of the model in distinguishing different clinical outcomes. An AUC value closer to 1.0 signifies strong predictive capability, while 0.5 indicates no discriminatory ability at all. By evaluating the AUC values generated by LR, SVM, and RF models, clinicians can identify which algorithm yields the most reliable predictions based on biomarker profiles for surgical patients.

## Immunohistochemistry

The protocol was performed as previously described (*Li et al., 2022*). In brief, the LEP and PTTG1 antibodies for IHC were purchased from Proteintech (Cat No: bs-0409R and bs-1881R). With deparaffinization for 15 min×3 in dimethylbenzene and routine hydration, the tissues were soaked in phosphate-buffered saline for 10 min and then performed high-pressure antigen retrieval (Tris-EDTA, pH = 9.0) for 2.5 min. After being treated with a 3% endogenous catalase blocker (ZSBIO, PV-6000) for 10 min, the tissues were incubated in goat serum (ZSBIO, ZLI-9022) for the blocking of nonspecific reaction and then incubated with primary antibody (LEP = 1:300 and PTTG1=1:300) at 4°C overnight. The next day, tissues were washed and incubated with goat anti-rabbit secondary antibody (ZSBIO, PV-9000) for 1 hr at room temperature, then washed and stained with DAB reagents (ZSBIO, ZLI-9018). Then, hematoxylin staining, 1% hydrochloric acid alcohol differentiation, ammonia water anti-blue, and neutral gum sealing.

The IHC results were evaluated by pathologists, and the staining extent was scored as 0–100%. The intensity score was defined as negative, low-expression, medium-expression, and high-expression, which were documented as 0, 1, 2, and 3, respectively. The final scores were calculated by the formula: *IHC score = Staining extent score × Staining intensity score*.

## Statistical methods

R software (v4.1.3) was used in this study. For quantitative variables, differences between the two groups and among multiple groups were analyzed by Wilcoxon's test and one-way analysis of variance

(ANOVA), respectively. For categorical variables, groups were compared by the use of a chi-square test. Survival curves were determined by the log-rank test. The clinicopathological features and levels of immune infiltration were conducted by Wilcoxon's test. A difference of p<0.05 indicated statistical significance unless specified otherwise.

## Results

Baseline characteristics were shown in *Table 1*. Of 329 RPLS patients, 88 in training cohort and 241 in validation cohort. No statistically significant differences were found in the age, sex, pathology, surgery times, tumor size, and multilocation between the two cohorts (p>0.05). In validation cohort, the IHC scores of LEP and PTTG1 were 1.62 (0.820) and 0.830 (0.75), respectively.

### Cell cycle, DNA damage and repair, and metabolism are dysregulated in RPLS

To reveal the general molecular features of RPLS compared to noncancerous adipose tissues, we first recruited eight RPLS patients and collected paired tumor and normal tissues for DEG analysis. A total of 1354 DEGs, 554 upregulated and 800 downregulated, were identified (*Figure 2A and B*). To assess the underlying pathways of RPLS, GSEAs were performed for those DEGs. We found that proliferation-associated pathways, such as mitotic spindle, E2F target, G2/M checkpoint, and separation of sister chromatids, were mainly enriched in tumors, while metabolism-related pathways, such as bile acid metabolism, heme and fatty acid metabolism, and integration of energy metabolism, were enriched in normal controls (*Figure 2C*).

Then, we collected another 80 samples to investigate the molecular heterogeneity of RPLS. Gene expression profiles showed 918 and 3244 genes associated with OS and DFS, respectively. Among 497 candidate genes associated with both OS and DFS, 83 of them also overlapped with DEGs (*Figure 2B*). Functional annotation (GO, KEGG, and enrichWP) demonstrated that cell cycle, DNA damage and repair, and metabolism-related pathways were significantly enriched (*Figure 2D*), suggesting these signaling pathways were dysregulated in RPLS.

### RPLS subgroups based on molecular features show different clinical outcomes

To evaluate heterogeneous molecular clustering characteristics in RPLS, ssGSEA emerged as a widely adopted method for computing the enrichment level of specific biological signaling pathways for each sample based on gene expression data. This aids in gaining insights into the overall activity level of signaling pathways. Here, we scored each sample on the dysregulated pathways by ssGSEA and divided RPLS patients into two subgroups (*Figure 3A*). Subgroup 1 (G1) showed better OS and DFS compared to subgroup 2 (G2) (*Figure 3B and C*). G1 displayed elevated ssGSEA scores associated with metabolism, whereas G2 exhibited heightened ssGSEA scores linked to cell cycle and DNA damage and repair (*Figure 3D*). These features suggested that effective monitoring of the prognosis of RPLS patients can be achieved based on the activation status of specific pathways. We also evaluated the clinical features and immune infiltration levels of those samples. The results showed that G1 had a lower aggressive pathological composition ratio, MDM2 and Ki67 expression, larger tumor size, and higher tumor microenvironment (TME) level compared to G2 (*Figure 3E, G, H*; *Figure 3—figure supplement 1A and B*). Surgery times for G1 were also tended to decrease (*Figure 3F*). Taken together, the above results indicated that RPLS subgroups based on molecular features showed distinct clinical features and clinical outcomes.

### A simplified RPLS classification strategy derived from RPLS G1/G2 subgroups

To explore representative biomarkers for different RPLS subgroups, we performed a DEG analysis between G1 and G2. There were 1258 genes downregulated, among which 112 of them indicated good prognosis (protective genes). Correspondingly, 754 genes were upregulated, and 28 of them indicated poor prognosis (aggressive genes) (*Figure 3—figure supplement 1C and D*). Enrichment analysis suggested that those DEGs were also associated with cell cycle regulation and metabolism (*Figure 3—figure supplement 1E*), which was consistent with previous results (*Figure 2C*).

**Table 1.** Baseline characteristics of training cohort and validation cohort.

| | Training cohort (N=80)* | Validation cohort (N=241) | p-Value |
|---|---|---|---|
| Age (years) | 56.34 (11.14)[†] | 55.11 (10.80) [†] | 0.384 |
| Sex | | | |
| Male | 37 (46.25) | 118 (48.96) | |
| Female | 43 (53.75) | 123 (51.04) | 0.674 |
| Pathology | | | |
| WDLS | 29 (36.25) | 75 (31.12) | |
| DDLS | 48 (60.00) | 144 (59.75) | |
| MLS and PLS | 3 (3.75) | 5 (2.07) | |
| NR | 0 (0) | 17 (7.06) | 0.078 |
| Surgery times [‡] | | | |
| 0–1 | 50 (62.50) | 143 (59.34) | |
| 2–3 | 21 (26.25) | 73 (30.29) | |
| 4–7 | 9 (11.25) | 24 (9.96) | |
| NR | 0 (0) | 1 (0.41) | 0.834 |
| Tumor size | | | |
| All | 18.65 (8.70)[†] | 16.90 (7.94)[†] | 0.101 |
| <18 cm | 40 (50.00) | 135 (56.02) | |
| >18 cm | 39 (48.75) | 92 (38.17) | |
| NR | 1 (1.25) | 14 (5.81) | 0.095 |
| Multilocation | | | |
| Yes | 52 (65.00) | 153 (63.49) | |
| No | 28 (35.00) | 74 (30.71) | |
| NR | 0 (0) | 14 (5.80) | 0.081 |
| MDM2 score | | | |
| 0 | 11 (13.75) | NA | |
| 1 | 12 (15.00) | NA | |
| 2 | 39 (48.75) | NA | |
| 3 | 5 (6.25) | NA | |
| 4 | 9 (11.25) | NA | |
| NR | 4 (5.00) | NA | NA |
| LEP score | NA | 1.62 (0.82)[†] | NA |
| LEP strength | | | |
| 0 | NA | 10 (4.15) | |
| 1 | NA | 39 (16.18) | |
| 2 | NA | 74 (30.71) | |
| 3 | NA | 115 (47.72) | |
| NR | NA | 3 (1.24) | NA |
| PTTG1 score | NA | 0.83 (0.75)[†] | NA |
| PTTG1 strength | | | |

*Table 1 continued on next page*

*Table 1 continued*

|  | Training cohort (N=80)* | Validation cohort (N=241) | p-Value |
|---|---|---|---|
| 0 | NA | 38 (15.77) | |
| 1 | NA | 100 (41.49) | |
| 2 | NA | 61 (25.31) | |
| 3 | NA | 39 (16.18) | |
| NR | NA | 3 (1.25) | NA |

*Clinical information missing in eight retroperitoneal liposarcoma (RPLS) patients (training cohort 2).

†The data is shown as mean (SD); other data is shown as number (%).

‡The definition of surgical times is the sum of current admission surgery and previous surgical resection DDLS, dedifferentiated liposarcoma; LEP, leptin; MDM2, mouse double minute 2; MLS, myxoid liposarcoma; NA, not applicable; NR, not reported; PLS, pleomorphic liposarcoma; PTTG1, pituitary tumor transforming gene 1; WDLS, well-differentiated liposarcoma.

To develop a simplified RPLS clustering based on DEGs, we adopted NMF and t-SNE for a reclassification of those patients. The results showed RPLS patients were also divided into two clusters (*Figure 4A* and *Figure 3—figure supplement 1F and G*). We then annotated the samples of two clusters by ssGSEA and found Cluster1 (C1) was related to metabolic processes, and Cluster2 (C2) was mainly related to the processes of cell cycle and DNA damage and repair (*Figure 4—figure supplement 1A*). Also, C1 showed better OS and DFS, lower pathological composition ratio and MDM2 expression, and fewer surgery times (*Figure 4B–F*). Lower Ki67 expression and larger tumor size were

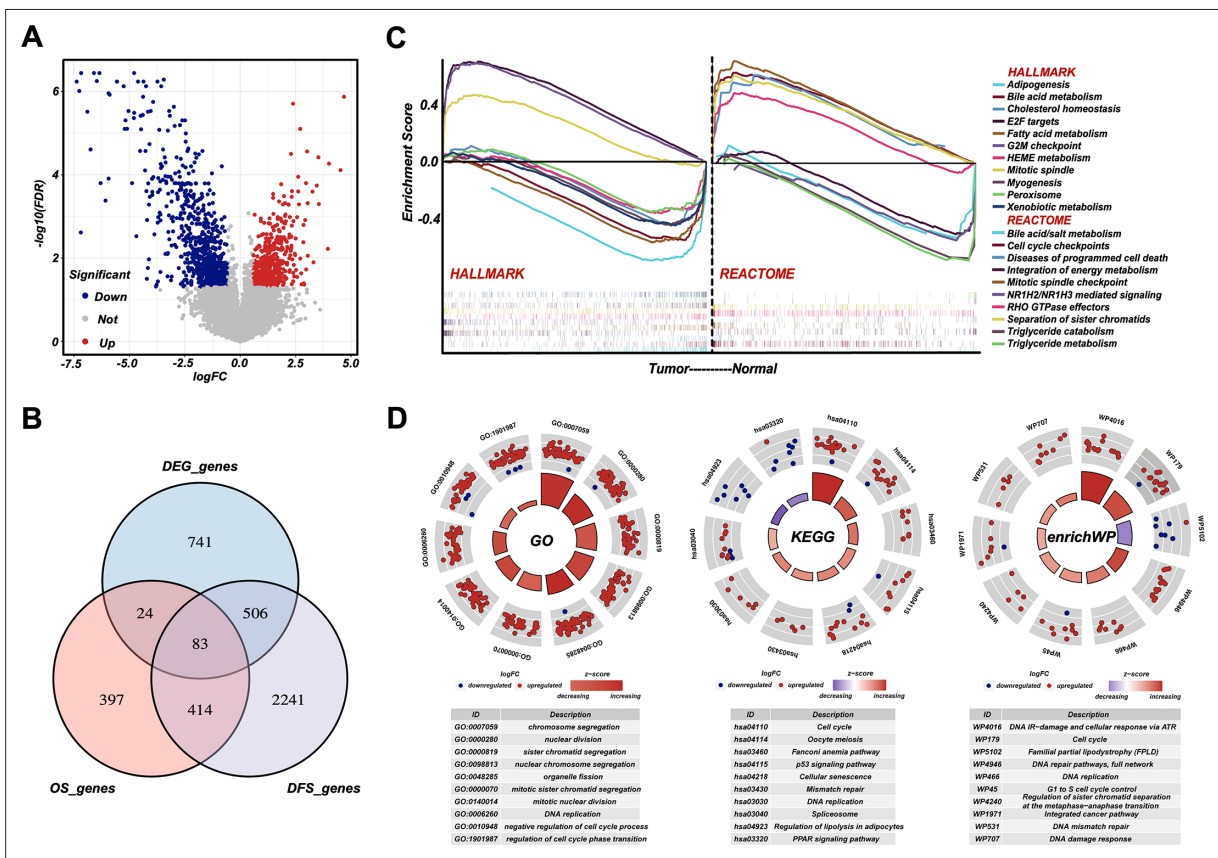

**Figure 2.** Cell cycle, DNA damage and repair, and metabolism are dysregulated in retroperitoneal liposarcoma (RPLS). Volcano plot of the differentially expressed genes (DEGs) in eight normal vs eight RPLS tissues (**A**). Venn diagram showing shared genes between DEGs and prognostic genes (**B**). Gene set enrichment analysis (GSEA) of RPLS tumors, including HALLMARK gene sets and REACTOME gene sets (**C**). Circular plots of the prognostic genes in Gene Ontology (GO), Kyoto Encyclopedia of Gene and Genomes (KEGG), and enrichWP (**D**).

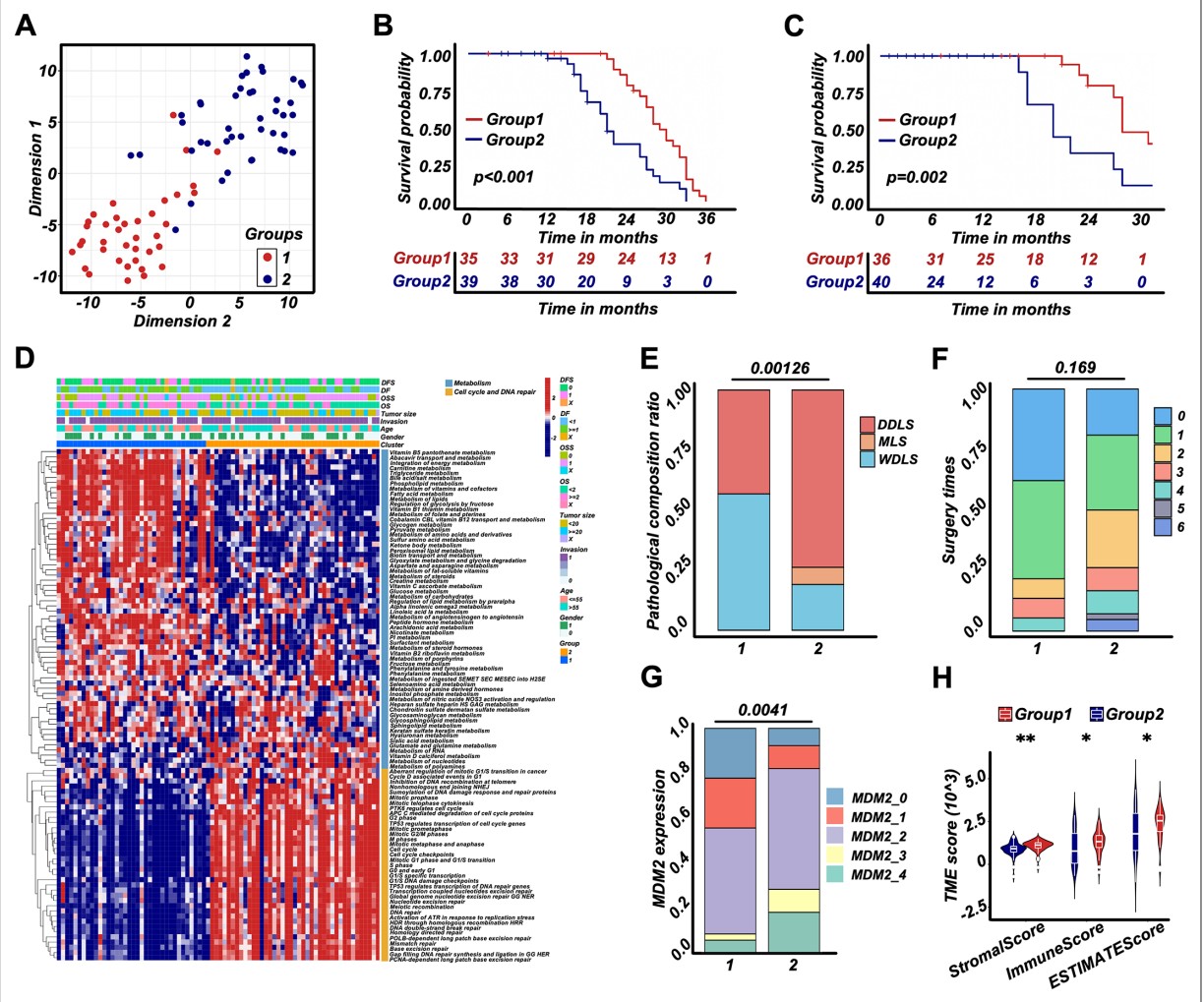

**Figure 3.** Retroperitoneal liposarcoma (RPLS) subgroups (**G1 and G2**) based on cell cycle, DNA damage and repair, and metabolism. t-Distributed stochastic neighbor embedding (t-SNE) exhibited the subgroups (**G1 and G2**) of RPLS (**A**). Survival cures of overall survival (OS) (**B**) and disease-free survival (DFS) (**C**) in G1 and G2. The hierarchical clustering heatmap of dysregulated pathways in G1 and G2 (**D**). Histograms revealed the difference in pathological composition ratio (**E**), surgery times (**F**), and MDM2 (**G**) in G1 and G2. Violin plot of the microenvironmental scores in G1 and G2 (**H**).

The online version of this article includes the following figure supplement(s) for figure 3:

**Figure supplement 1.** Clinical features and re-classification of retroperitoneal liposarcoma (RPLS) subgroups (G1 and G2).

observed in C2 (*Figure 4—figure supplement 1B and C*). Interestingly, the biological annotations of the C1/C2 classification were greatly consistent with G1/G2. Therefore, a simplified RPLS classification strategy derived from RPLS subgroups was provisionally established.

## Development of a dichotomous RPLS classification model

For NMF classification of RPLS patients, LEP and PTTG1 were identified as representative biomarkers of C1 and C2, respectively (*Figure 5A*). The selection criteria for identifying LEP and PTTG1 as biomarkers involved selecting prognostic genes that were highly expressed in C1 and C2, respectively, and achieved the highest AUC value in distinguishing the two RPLS groups. We aimed to replicate the RPLS classification of C1 and C2 by integrating these two biomarkers with the assistance of machine learning algorithms, and this two-gene panel achieved promising results (logistic, AUC = 0.995; SVM, AUC = 0.997; RF, AUC = 1.000; *Figure 5B*). Also, a linear negative correlation between LEP and PTTG1 expression was detected (*Figure 5C*). Considering the enhanced interpretability and generalization of linear models, we adopted the results of LR for subsequent analysis (risk values = 2.182 × PTTG1−2.204×LEP). The patients marked as high-risk (Cluster_H) exhibited worse

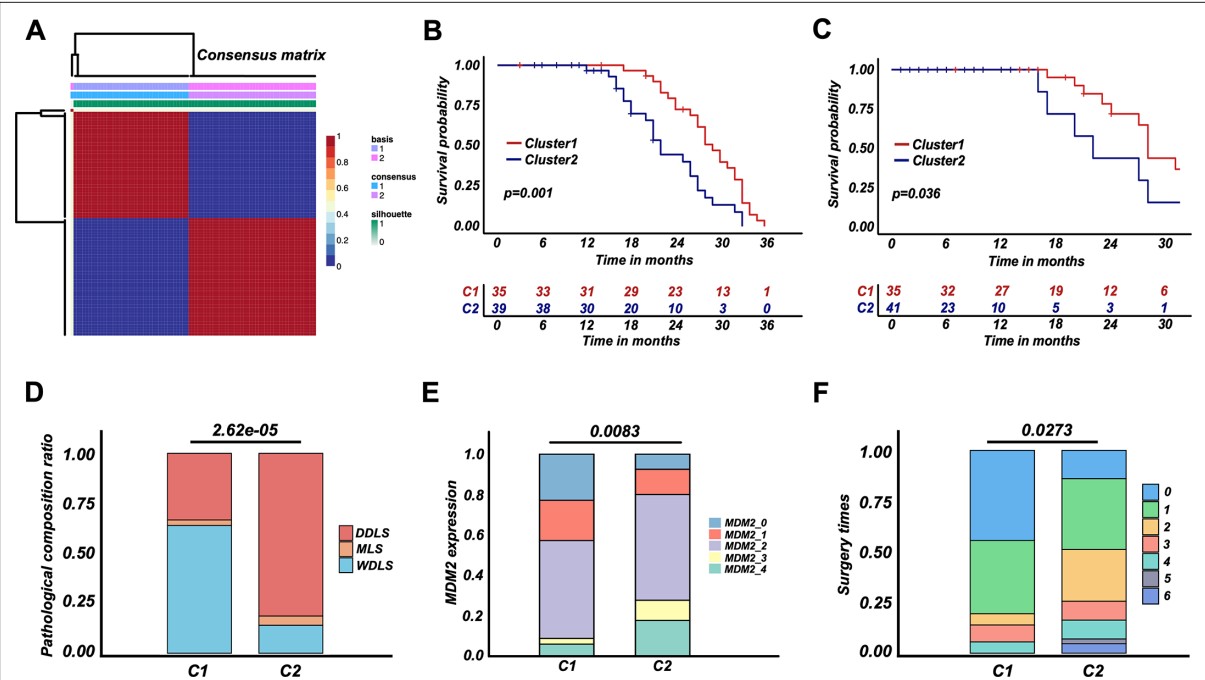

**Figure 4.** Retroperitoneal liposarcoma (RPLS) classification strategy (**C1 and C2**) derived from RPLS subgroups. Nonnegative matrix factorization (NMF) for a reclassification of training cohort 1 (**C1 and C2**) (**A**). Survival cures of overall survival (OS) (**B**) and disease-free survival (DFS) (**C**) in C1 and C2. Histograms revealed the difference in pathological composition ratio (**D**), MDM2 (**E**), and surgery times (**F**) in C1 and C2.

The online version of this article includes the following figure supplement(s) for figure 4:

**Figure supplement 1.** Dysregulated pathways and clinical features of retroperitoneal liposarcoma (RPLS) clusters and high-/low-risk groups.

The hierarchical clustering heatmap of dysregulated pathways in C1 and C2 (**A**). The difference ofin Ki67 (**B**) and tumor size (**C**) in C1 and C2. The hierarchical clustering heatmap of dysregulated pathways in high- and low-risk groups (**D**). The difference ofin tumor size (**E**) and Ki67 (**F**) in high- and low-risk groups.

OS and DFS than those marked as low-risk (Cluster_L) (*Figure 5D and E*). Dysregulated pathways, such as DNA repair and cell cycle regulation, were enriched in Cluster_H (*Figure 5F*; *Figure 4— figure supplement 1D*), and Cluster_H presented more aggressive pathological composition ratio, higher MDM2 levels, and marginally increased in surgery times than Cluster_L (*Figure 5G–I*). Similarly, the Cluster_H showed higher Ki67 levels and smaller tumor sizes (*Figure 4—figure supplement 1E and F*). Moreover, a Sankey diagram was drawn to show the correlation among G1/G2, C1/C2, and Cluster_L/H. Cluster_L/H was well matched to C1/C2 and G1/G2, suggesting LEP and PTTG1 were promising biomarkers for a dichotomous RPLS classification (*Figure 5J*). To ensure the broader applicability of LEP and PTTG1 as classification biomarkers, we performed an independent validation using an external liposarcoma cohort (GSE30929). This dataset was selected due to its relevance to RPLS (N=63, 45%) and the availability of distant recurrence-free survival (DRFS) outcomes, aligning with the clinical focus of our study. Applying our established LR, we found that the high-risk (V) group exhibited significantly worse DRFS compared to the low-risk (V) group (*Figure 5—figure supplement 1A and B*), and the high-risk group (V) demonstrated a higher proportion of high-grade histology (*Figure 5—figure supplement 1C and D*), consistent with our previous findings. These results validate the robustness and generalizability of our risk stratification model across distinct liposarcoma cohorts. The external dataset's alignment with our findings underscores the potential of LEP and PTTG1 as reproducible biomarkers for prognosis and therapeutic stratification in liposarcoma.

## Validation of the dichotomous RPLS classification in another 241 RPLS patients

To validate LEP and PTTG1 as biomarkers for a dichotomous RPLS classification, we performed IHC staining of two biomarkers in validation cohort. The representative images of LEP and PTTG1 with

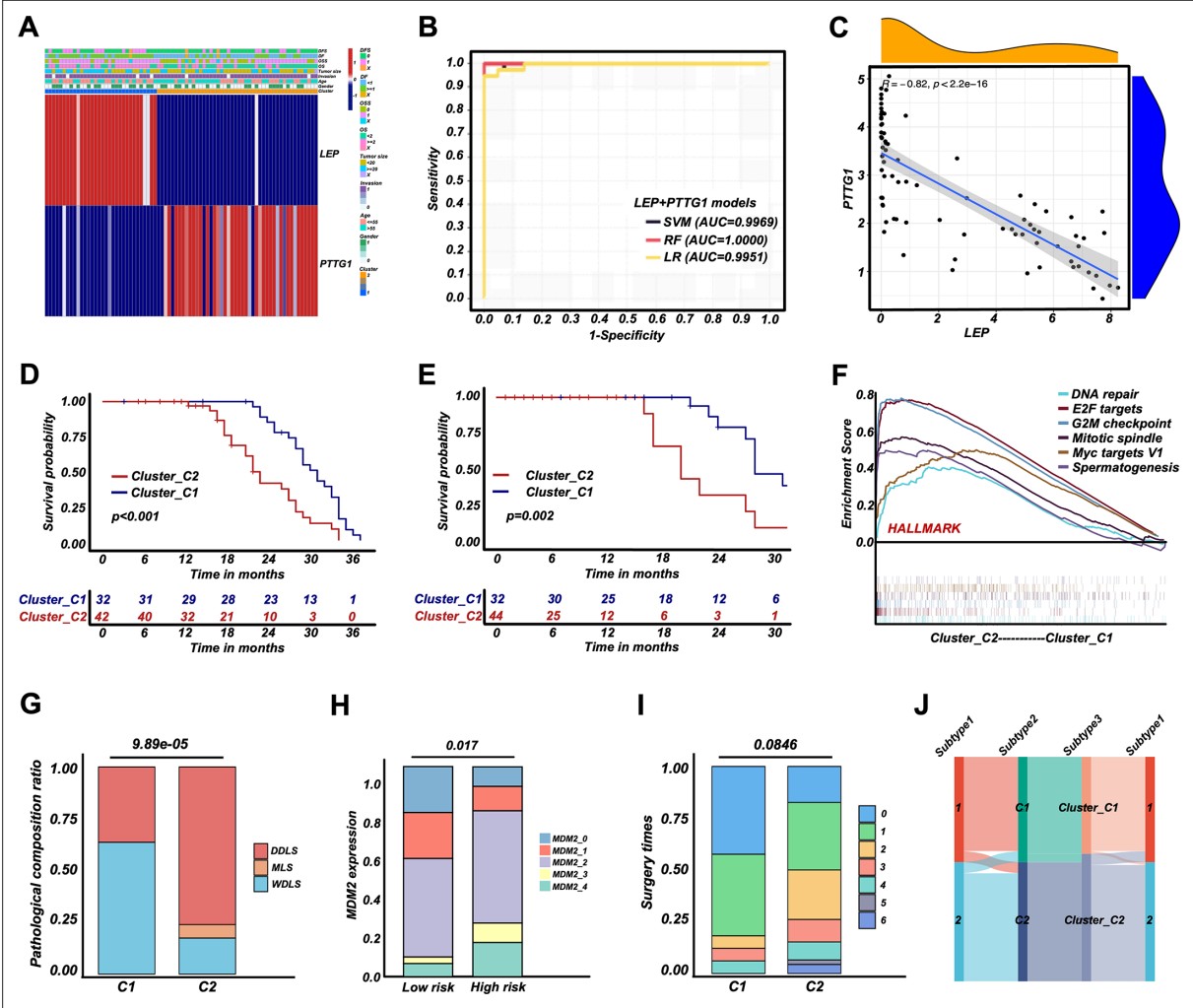

**Figure 5.** Retroperitoneal liposarcoma (RPLS) dichotomous classification (Cluster_C1 and Cluster_C2) derived from RPLS clusters. Heatmap of biomarkers identified (LEP and PTTG1) in C1 and C2 (**A**). ROC curves of the machine learning models to identify C1 and C2 (**B**). Correlation between LEP and PTTG1 expression (**C**). Survival curves of overall survival (OS) (**D**) and disease-free survival (DFS) (**E**) in Cluster_C1 (low-risk) and Cluster_C2 (high-risk) groups. Gene set enrichment analysis (GSEA) of HALLMARK gene sets in Cluster_C1 and Cluster_C2 (**F**). Histograms revealed the difference in pathological composition ratio (**G**), MDM2 level (**H**), and surgery times (**I**) in Cluster_C1 and Cluster_C2. Sankey diagram indicated the correlation among G1/G2, C1/C2, and Cluster_C1/Cluster_C2 (**J**).

The online version of this article includes the following figure supplement(s) for figure 5:

**Figure supplement 1.** Validation of LEP+PTTG1 model in an external liposarcoma cohort.

different expression levels were shown in *Figure 6A and B*. The IHC scores were integrated with the previously fitted coefficients to evaluate the prognosis of RPLS patients (risk values = 2.182×PTTG1$_{IHC}$-2.204×LEP$_{IHC}$). The cutoff value of validation cohort is the median of the risk value. The high-risk group had worse OS and DFS (*Figure 6C and D*), along with more surgery times and more aggressive pathological composition ratio (*Figure 7A and B*), but the difference in tumor size was not observed between the two groups (*Figure 7C*). Then, we constructed visual nomograms for a precise survival prediction of RPLS patients by combining the risk score with clinical features. The predictive abilities of the 1-, 2-, and 3-year OS (*Figure 7D–F*) and DFS (*Figure 7—figure supplement 1A–C*) were 0.743–0.788. Together, we proposed a simple and clinically applicable molecular classification strategy for RPLS patients.

## Discussion

Here, we divided RPLS patients into two subgroups based on cell cycle, DNA damage and repair, and metabolism-related pathways. G1 was annotated as metabolism-active, which exhibited high ssGSEA

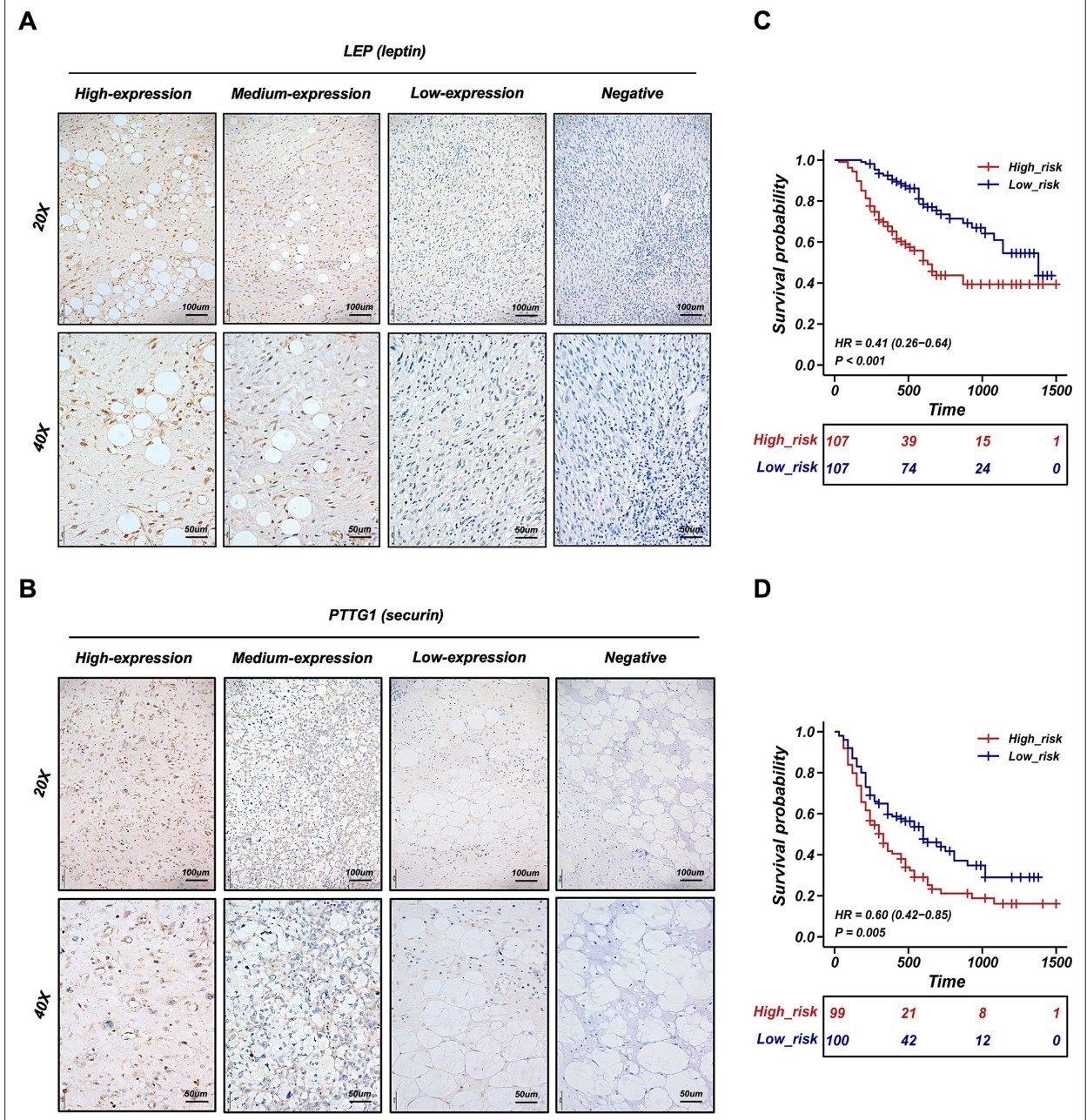

**Figure 6.** Validation of the retroperitoneal liposarcoma (RPLS) dichotomous classification in another 241 RPLS cohort. Representative immunohistochemistry (IHC) staining images of LEP (**A**) and PTTG1 (**B**). Survival curves of overall survival (OS) (**C**) and disease-free survival (DFS) (**D**) in high-risk and low-risk groups.

scores on metabolism-associated pathways, while G2 showed high ssGSEA scores on cell cycle and DNA damage and repair with a high Ki67 and MDM2 level.

G2 had more aggressive molecular features and worse clinical outcomes compared to G1, in accordance with previously reported tumor classification (*Lindskrog et al., 2021*; *Yu et al., 2021*; *Zhang et al., 2022*). In fact, the *Cancer Genome Atlas Research Network. Electronic address: elizabeth. demicco@sinaihealthsystem.ca and Cancer Genome Atlas Research Network, 2017*, has integrated SCNA and DNA methylation to divide dedifferentiated liposarcoma into two subtypes (S1 and S2), the unfavorable cluster was characterized as JUN amplified (an oncogene that promotes proliferation and metastasis) and lower inferred fraction of immature dendritic cells. However, the patients in the *Cancer Genome Atlas Research Network. Electronic address: elizabeth.demicco@*

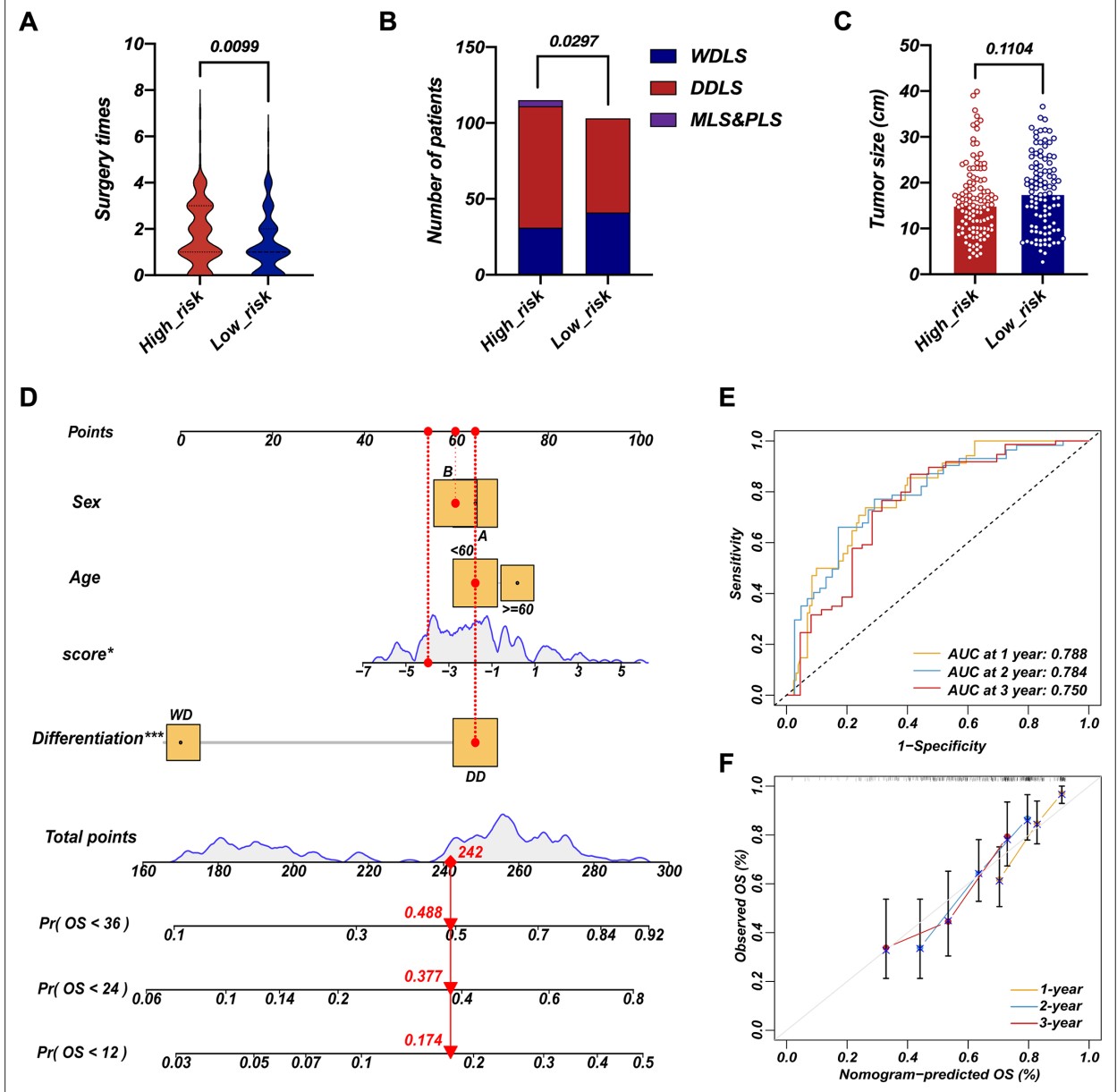

**Figure 7.** Survival nomogram of LEP+PTTG1 model in validation cohort. The difference in surgery times (**A**), pathological composition (**B**), and surgery times (**C**) in high-risk and low-risk groups. Nomograms for overall survival (OS) were developed in REASR cohort with four factors: sex, age, risk score, and differentiation (**D**). ROC curves of 1-, 2-, and 3-year OS in validation cohort (**E**). Calibration curves of predicting 1-, 2-, and 3-year OS in validation cohort (**F**).

The online version of this article includes the following figure supplement(s) for figure 7:

**Figure supplement 1.** Survival nomogram of LEP+PTTG1 model in validation cohort.

*sinaihealthsystem.ca and Cancer Genome Atlas Research Network, 2017*, were of complex origin (mixed limbs, trunk, and retroperitoneum), providing limited guidance for RPLS molecular classification. Here, we reported the first clinically applicable RPLS molecular classification based on RNA sequencing and IHC validation cohorts.

To facilitate the clinical application, we constructed a simplified RPLS molecular classification derived from the original cell cycle/metabolism subgroups. By the NMF algorithm, we identified LEP and PTTG1 as representative biomarkers for each subtype. A model based on IHC staining of LEP and PTTG1 successfully approximated the original dichotomous RPLS classification in biological features and survival outcomes. LEP is an important regulator of basal metabolism and food intake, which

is considered a linkage between metabolism and the immune system (*Jiménez-Cortegana et al., 2021*). Although LEP-based targeting therapies have not yet been fully applied, LEP has already been identified as a potent metabolic reprogramming agent to support antitumor responses in aggressive melanomas (*Waldman et al., 2020*; *de la Cruz-Merino et al., 2019*; *Rivadeneira et al., 2019*). In addition, LEP improves the immunotherapeutic effects by regulating innate and adaptive immune responses via increasing the cytotoxicity of NK cells (*Francisco et al., 2018*), stimulating the proliferation of T/B cells (*Francisco et al., 2018*; *Bernotiene et al., 2006*), and activating DC cells (*Hu et al., 2019*). Those reported roles of LEP provided a good mechanism explanation on the features of metabolism pathway-enriched, better prognosis, and higher TME level of metabolism subgroup (LEP+). In contrast, PTTG1 acts as a regulator of sister chromatid separation during cell division under physiological conditions (*Zou et al., 1999*), which is closely linked to genetic instability, aneuploidy, tumor progression, invasion, and metastasis (*Heaney et al., 2000*; *Ramaswamy et al., 2003*; *Kim et al., 2005*; *Yu et al., 2003*; *Teveroni et al., 2021*; *Romero et al., 2001*). PTTG1 also regulates the cell cycle and the transactivation of growth factors as an initiator and promoter of tumorigenesis (*Zou et al., 1999*; *Mora-Santos et al., 2013*; *McCabe et al., 2002*; *Ishikawa et al., 2001*; *Hamid et al., 2005*). Overexpressing PTTG1 was correlated with worse prognosis in tumors, such as ovarian cancer (*Parte et al., 2019*), cervical cancer (*Guo et al., 2019b*), renal cell carcinoma (*Tian et al., 2022*), and colorectal cancer (*Heaney et al., 2000*). Therefore, the biological functions of PTTG1 provided a good mechanism explanation of the pathway-enriched cell cycle/DNA damage and repair-associated, worse prognosis, and more aggressive pathological composition ratio in the cell cycle subgroup (PTTG1+).

## Conclusion

Our study presented a comprehensive gene expression landscape of RPLS, revealing distinct molecular features. Through categorizing RPLS into metabolism and cell cycle subtypes and identifying key biomarkers LEP and PTTG1, we established a dichotomous classification system verified by IHC assays. This innovative approach enables precise guidance for surgeons in adjusting treatment strategies for patients with histologically favorable but prognostically challenging RPLS cases, thereby advancing the implementation of precision medicine in guiding surgical interventions for RPLS.

## Data anonymization

All participant identifiers were replaced with unique IDs during data collection and analysis.

## Acknowledgements

This work was supported by grants from the Beijing Municipal Science and Technology Project (Z191100006619081), National Natural Science Foundation of China (82073390), and Young Elite Scientists Sponsorship Program (2023QNRC001). The study sponsors had no role in the design and preparation of this manuscript.

## Additional information

### Funding

| Funder | Grant reference number | Author |
|---|---|---|
| Beijing Municipal Science and Technology Commission, Adminitrative Commission of Zhongguancun Science Park | Z191100006619081 | Li Min |
| National Natural Science Foundation of China | 82073390 | Li Min |
| Young Elite Scientists Sponsorship Program | 2023QNRC001 | Mengmeng Xiao |

| Funder | Grant reference number | Author |
|--------|------------------------|--------|

The funders had no role in study design, data collection and interpretation, or the decision to submit the work for publication.

## Author contributions

Mengmeng Xiao, Data curation, Formal analysis, Methodology, Writing - original draft, Writing – review and editing; Xiangji Li, Data curation, Formal analysis, Investigation, Methodology, Writing - original draft; Fanqin Bu, Conceptualization, Data curation, Formal analysis, Investigation, Methodology, Writing – review and editing; Shixiang Ma, Formal analysis, Methodology; Xiaohan Yang, Investigation, Methodology; Jun Chen, Formal analysis; Yu Zhao, Ferdinando Cananzi, Investigation; Chenghua Luo, Conceptualization, Supervision, Methodology, Writing – review and editing; Li Min, Conceptualization, Resources, Supervision, Writing – review and editing

## Author ORCIDs

Chenghua Luo (ID) https://orcid.org/0000-0002-5939-7888
Li Min (ID) https://orcid.org/0000-0001-9595-5536

## Ethics

Clinical trial registration Unique identifying number or registration ID: NCT03838718.
Specimens of RPLS were obtained from Peking University International Hospital. The study protocol was approved by the Ethics Committee of Peking University International Hospital, Peking University Health Science Center (WA2020RW29) and conducted in accordance with Helsinki Declaration. All patients signed the informed consent.

Reviewer #1 (Public review): https://doi.org/10.7554/eLife.100887.3.sa1
Author response https://doi.org/10.7554/eLife.100887.3.sa2

# Additional files

## Supplementary files

Supplementary file 1. The detailed clinicopathological characteristics of the training cohort 1.
Supplementary file 2. The RNA-seq data of the training cohort 2.
Supplementary file 3. The detailed clinicopathological characteristics of the validation cohort.
Supplementary file 4. The RNA-seq data of the training cohort 1.
MDAR checklist

## Data availability

Data supporting the conclusions of this article are presented within the article and its supplementary files. Source data of Figure 2–5, Figure 3—figure supplement 1 and Figure 4—figure supplement 1 are available in Supplementary files 1, 2 and 4. Source data of Figure 6–7 and Figure 7—figure supplement 1 are available in Supplementary files 3. Source data of Figure 5—figure supplement 1 is available in GSE30929. The sequencing data (Supplementary files 1–4) of retroperitoneal liposarcoma have been deposited at the Open Archive for Miscellaneous Data (OMIX) database of China National Center for Bioinformation (CNCB) under the accession number OMIX002786. The data of the external validation cohort (GSE30929) was downloaded from GEO (https://www.ncbi.nlm.nih.gov/geo/). This paper does not report the original code.

The following previously published datasets were used:

| Author(s) | Year | Dataset title | Dataset URL | Database and Identifier |
|-----------|------|---------------|-------------|-------------------------|
| Xiao MM | 2023 | Whole genome sequencing of retroperitoneal liposarcoma | https://ngdc.cncb.ac.cn/omix/release/OMIX002786 | CNCB, OMIX002786 |

*Continued on next page*

*Continued*

| Author(s) | Year | Dataset title | Dataset URL | Database and Identifier |
|---|---|---|---|---|
| Gobble RM, Qin LX, Brill ER, Angeles CV | 2011 | Whole-transcript expression data for liposarcoma | https://www.ncbi.nlm.nih.gov/geo/query/acc.cgi?acc=GSE30929 | NCBI Gene Expression Omnibus, GSE30929 |
| Xiao MM | 2023 | Whole genome sequencing of retroperitoneal liposarcoma | https://ngdc.cncb.ac.cn/omix/release/PRJCA014351 | CNCB, PRJCA014351 |

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
