## [Editor Report · eLife Assessment]

This study presents a **valuable** and simplified classification system for predicting clinical outcomes of RPLS patients. The data were collected and analyzed using **solid** and validated methodology and can be used as a starting point for personalized treatment of RPLS. The work will be of interest to scientists working in the field of RPLS.

---

## [Referee Report · Reviewer #1 (Public review)]

Summary:

In this study, Xiao et al. conducted a comprehensive analysis of retroperitoneal liposarcoma (RPLS) by classifying patients into two distinct molecular subgroups based on whole transcriptome sequencing data from 88 cases. The G1 subgroup demonstrated a metabolic activation signature, whereas the G2 subgroup was characterized by enhanced cell cycle regulation and DNA damage repair pathways. Notably, the G2 subgroup exhibited more aggressive molecular profiles and poorer clinical prognosis compared to the G1 subgroup. Through the application of machine learning algorithms, the authors established a streamlined classification system, identifying LEP and PTTG1 as pivotal molecular biomarkers for differentiating between these two RPLS subgroups. The manuscript presents a well-structured and methodologically sound study, with particular significance attributed to its substantial sample size and the development of a clinically applicable classification framework. This innovative model holds considerable promise for advancing personalized treatment strategies and improving clinical outcomes for RPLS patients.

Comments on revisions:

The authors have adequately addressed all my concerns, and I have no further comments.

---

## [Author Response]

The following is the authors’ response to the original reviews

**Public Reviews:**

**Reviewer #1 (Public review):**
Summary:In this study, Xiao et al. classified retroperitoneal liposarcoma (RPLS) patients into two subgroups based on whole transcriptome sequencing of 88 patients. The G1 group was characterized by active metabolism, while the G2 group exhibited high scores in cell cycle regulation and DNA damage repair. The G2 group also displayed more aggressive molecular features and had worse clinical outcomes compared to G1. Using a machine learning model, the authors simplified the classification system, identifying LEP and PTTG1 as the key molecular markers distinguishing the two RPLS subgroups. Finally, they validated these markers in a larger cohort of 241 RPLS patients using immunohistochemistry. Overall, the manuscript is clear and well-organized, with its significance rooted in the large sample size and the development of a classification method.

Thank you for your positive assessment of our study on classifying RPLS patients based on whole transcriptome sequencing. We appreciate your recognition of the distinct characteristics of the G1 and G2 groups, as well as the significance of our simplified classification system and the identification of LEP and PTTG1 as key molecular markers. Your acknowledgment of the clarity and organization of our manuscript, along with the importance of the large sample size, is greatly appreciated. We will continue to refine our work based on your feedback as we prepare for resubmission.

Weakness:(1) While the authors suggest that LEP and PTTG1 serve as molecular markers for the two RPLS groups, the process through which these genes were selected remains unclear. The authors should provide a detailed explanation of the selection process.

The selection criteria for identifying LEP and PTTG1 as biomarkers involved selecting prognostic genes that were highly expressed in C1 and C2, respectively, and achieved the highest AUC value in distinguishing the two RPLS groups (Page17 lines 288-290).

(2) To ensure the broader applicability of LEP and PTTG1 as classification markers, the authors should validate their findings in one or two external datasets.

We sincerely appreciate your insightful suggestion regarding the external validation of LEP and PTTG1 as classification biomarkers. To address this concern, we performed an independent validation using an external liposarcoma cohort (GSE30929; Page 6, Lines 104-105), which comprises 140 primary liposarcoma samples with annotated clinicopathological and survival data. This dataset was selected due to its relevance to RPLS (N=63, 45%) and the availability of distant recurrence-free survival (DRFS) outcomes, aligning with the clinical focus of our study.

Applying our previously established prognostic model (Risk value = 2.182 × PTTG1 - 2.204 × LEP) to this cohort, we stratified patients into high- and low-risk groups using the median risk score as the cutoff. Consistent with our original findings, the high-risk group exhibited significantly worse DRFS compared to the low-risk group. The ROC curves based on the 1-, 3-, 5-year survival status of patients demonstrated that this model can effectively predict patient DRFS (log-rank P < 0.001, Figure S3A-B). Furthermore, the high-risk group demonstrated a higher proportion of high-grade histology (P < 0.001, Fisher’s exact test, Figure S3C-D).

These results validate the robustness and generalizability of our risk stratification model across distinct liposarcoma cohorts. The external dataset’s alignment with our findings underscores the potential of LEP and PTTG1 as reproducible biomarkers for prognosis and therapeutic stratification in liposarcoma. We have incorporated these validation results into the revised manuscript (Page 18, Lines 305-315) to strengthen the clinical applicability of our conclusions.

(3) Since molecular subtyping is often used to guide personalized treatment strategies, it is recommended that the authors evaluate therapeutic responses in the two distinct groups. Additionally, they should validate these predictions using cell lines or primary cells.

We sincerely appreciate your insightful comments and suggestions regarding the evaluation of therapeutic responses and the validation of our predictions using cell lines or primary cells. We would like to address these points in detail below:

(1) Purpose of the PTTG1- and LEP-based RPLS Classification Model

The primary objective of our study was to develop a molecular subtyping model based on PTTG1 and LEP to guide personalized treatment strategies for patients with RPLS, particularly those classified as low-grade by traditional histopathological criteria but exhibiting poor prognosis. This subgroup of patients may benefit from more aggressive surgical resection, which is a potentially curative approach for RPLS. Our model aims to identify these high-risk patients to ensure complete tumor resection, thereby improving their clinical outcomes.

(2) Therapeutic Response Evaluation in Distinct Groups

In both our validation cohort and external validation cohort, surgical resection was the primary treatment modality for RPLS. After stratifying patients using our model, we observed significant differences in surgical outcomes between the two groups: the high-risk group exhibited poor prognosis, while the low-risk group showed favorable outcomes (Figure 5D-E and Figure S3A-B). Importantly, our model successfully identified low-grade histopathological cases with poor prognosis, who might otherwise be undertreated (Figure 5G-I and Figure S3C-D). By advocating for more thorough surgical resection in these high-risk patients, we aim to improve their prognosis. This achievement aligns with the primary goal of our study, which is to provide a molecular tool for personalized treatment guidance.

(3) Future Validation and Functional Exploration of PTTG1 and LEP

Our study has identified PTTG1 and LEP as key biomarkers for RPLS classification, and we recognize the urgent need to elucidate their molecular functions in RPLS pathogenesis. Here, we are pleased to report that we have already initiated cellular and animal experiments to investigate the roles of PTTG1 and LEP in RPLS. These experiments aim to validate our predictions and explore the underlying mechanisms by which these biomarkers contribute to tumor behavior and treatment response. We anticipate that the results of these studies will provide further mechanistic insights and will be submitted for publication in a suitable journal in the near future.

**Reviewer #2 (Public review):**
Surgical resection remains the most effective treatment for retroperitoneal liposarcoma. However, postoperative recurrence is very common and is considered the main cause of disease-related death. Considering the importance and effectiveness of precision medicine, the identification of molecular characteristics is particularly important for the prognosis assessment and individualized treatment of RPLS. In this work, the authors described the gene expression map of RPLS and illustrated an innovative strategy of molecular classification. Through the pathway enrichment of differentially expressed genes, characteristic abnormal biological processes were identified, and RPLS patients were simply categorized based on the two major abnormal biological processes. Subsequently, the classification strategy was further simplified through nonnegative matrix factorization. The authors finally narrowed the classification indicators to two characteristic molecules LEP and PTTG1, and constructed novel molecular prognosis models that presented obviously a great area under the curve. A relatively interpretable logistic regression model was selected to obtain the risk scoring formula, and its clinical relevance and prognostic evaluation efficiency were verified by immunohistochemistry. Recently, prognostic model construction has been a hot topic in the field of oncology. The interesting point of this study is that it effectively screened characteristic molecules and practically simplified the typing strategy on the basis of ensuring high matching clinical relevance. Overall, the study is well-designed and will serve as a valuable resource for RPLS research.

Thank you for your insightful feedback on our manuscript. We appreciate your recognition of the importance of precision medicine and molecular characteristics in improving prognosis and individualized treatment for RPLS.

We are pleased that you found our gene expression mapping and innovative molecular classification strategy valuable. Your positive remarks on our pathway enrichment analysis and the categorization of RPLS patients based on abnormal biological processes affirm our approach.

We are also grateful for your acknowledgment of our focus on the characteristic molecules LEP and PTTG1, as well as the development of novel molecular prognosis models with significant predictive capability.